# Non-STEMI vs. STEMI Cardiogenic Shock: Clinical Profile and Long-Term Outcomes

**DOI:** 10.3390/jcm11123558

**Published:** 2022-06-20

**Authors:** María José Martínez, Ferran Rueda, Carlos Labata, Teresa Oliveras, Santiago Montero, Marc Ferrer, Nabil El Ouaddi, Jordi Serra, Josep Lupón, Antoni Bayés-Genís, Cosme García-García

**Affiliations:** 1Cardiology Department, Heart Institute, Hospital Universitari Germans Trias i Pujol, 08916 Badalona, Spain; mj.martinez.membrive@gmail.com (M.J.M.); fruedasobella@hotmail.com (F.R.); clabata@hotmail.com (C.L.); 3aoliveras@gmail.com (T.O.); monteroaradas@gmail.com (S.M.); mafema1986@hotmail.com (M.F.); elouaddi@hotmail.com (N.E.O.); jserraflores.germanstrias@gencat.cat (J.S.); jlupon.germanstrias@gencat.cat (J.L.); abayesgenis@gmail.com (A.B.-G.); 2Cardiology Department, Hospital del Mar, 08003 Barcelona, Spain; 3PhD Program, Department of Medicine, Autonomous University of Barcelona, 08193 Barcelona, Spain; 4CIBER Enfermedades Cardiovasculares (CIBERCV), 28029 Madrid, Spain; 5Department of Medicine, Autonomous University of Barcelona, 08193 Barcelona, Spain

**Keywords:** STEMI, NSTEMI, cardiogenic shock, prognosis

## Abstract

Cardiogenic shock (CS) is a severe complication of acute myocardial infarction (AMI). In AMI-CS, the ST segment deviation on ECG may be elevated (STEMI-CS) or non-elevated (NSTEMI-CS), which may influence prognosis. Our aim was to analyze the clinical profile, acute-phase prognosis, and long-term outcomes of CS relative to the ST pattern on admission. In a prospective registry of 4647 AMI patients admitted to the intensive cardiac care unit of a university hospital between 2010 and 2019, we compared the clinical characteristics, 30-days case fatality, and long-term outcomes of AMI-CS, based on the presence of ST-segment deviation. AMI-CS developed in 239 (5.1%) patients (26.4% women): 190 (79.5%) STEMI-CS and 49 (20.5%) NSTEMI-CS. The mean age was 69.7 years. The STEMI-CS patients had larger infarcts and more mechanical complications than the NSTEMI-CS patients. The NSTEMI-CS patients had a greater prevalence of hypertension, diabetes, peripheral vascular disease, previous cardiovascular comorbidities, three-vessel disease, and left main disease than the STEMI-CS patients. The STEMI-CS patients had higher 30-day mortality than the NSTEMI-CS (59.5% vs. 36.7%; *p* = 0.004), even after multivariable adjustment (HR 1.91; 95% CI 1.16–3.14), but no differences in mortality were observed at 3 years. In conclusion, the 30-day case-fatality is higher in STEMI-CS, but the long-term outcome is similar in both groups.

## 1. Introduction

Acute myocardial infarction (AMI) is the main cause of cardiogenic shock (CS) and is associated with higher mortality, despite current intensive therapies and coronary revascularization [1,2,3]. AMI is classified into ST-elevation (STEMI) or non-ST elevation (NSTEMI), according to the patient’s electrocardiogram (ECG) at admission. The pathophysiology of these two groups is not the same, and this classification helps in making decisions on therapeutic management in the acute phase.

Rigorous studies have compared mortality between STEMI and NSTEMI patients and concluded that STEMI has higher in-hospital death and NSTEMI has a worse long-term prognosis [4,5,6]. With regards to AMI complicated with CS (AMI-CS), some studies have suggested that the patients with STEMI have a higher prevalence of CS [7], though information is scarce about the prevalence, in-hospital case-fatality, and the long-term prognosis of CS depending on the type of AMI. The predictors of in-hospital survival that have been described in the literature on CS include older age, prior stroke, glucose and creatinine at admission, abnormal coronary flow after percutaneous intervention, and elevated serum lactate [3,8], yet information relative to the prognosis based on the admission ECG pattern is not well established. Accordingly, the aim of the current study was to analyze the differences in clinical profiles, in-hospital case-fatality, and long-term prognosis among patients with AMI-CS due to acute AMI relative to the electrocardiographic pattern at admission (STEMI-CS vs. NSTEMI-CS).

## 2. Materials and Methods

### 2.1. Study Population and Data Collection

We prospectively studied all of the consecutive patients with AMI-CS who were admitted to the intensive cardiac care unit (ICCU) of a single tertiary university hospital from 1 January 2010 to 31 December 2019. The center serves approximately 850,000 inhabitants from the north metropolitan area of Barcelona (Spain) and had a 24/7 catheterization laboratory and cardiac surgery availability during the study period. All of the patients received standard care following the recommendations from the current European Society of Cardiology Guidelines [9,10].

CS was defined as hypotension for at least 30 min (systolic blood pressure (SBP) <90 mmHg or catecholamines to maintain SBP > 90 mmHg), pulmonary or venous congestion, and signs of hypoperfusion (altered mental status, cold periphery, oliguria <0.5 mL/kg/h for the previous 6 h, or blood lactate >2 mmol/L) [11,12]. In the patients in whom pulmonary artery catheterization was performed, a cardiac index ≤2.2 L/min/m^2^ and pulmonary capillary wedge pressure ≥15 mmHg were required [3,11,13]. The diagnosis of AMI was established according to the current universal definition of myocardial infarction during the study [14,15,16]. This registry only included type 1 AMI. AMI was classified as STEMI when a persistent (e.g., >20 min) new ST elevation at the J point ≥0.1 mV in two contiguous leads (in V2–3 ≥ 0.2 mV in men and ≥0.15 mV in women) or new onset left bundle branch block was present on the ECG. AMI presenting with other electrocardiographic patterns was classified as NSTEMI.

The data were prospectively collected at admission, including the baseline characteristics, clinical presentation, laboratory values, coronary anatomy, revascularization features, procedures and medical therapies, and in-hospital complications. The left ventricular ejection fraction (LVEF) was assessed at admission and discharge (or before death) with echocardiography, using the Simpson method. The follow-up was performed by the investigators at three specified time points (1, 3, and 5 years after admission) by telephone contact and electronic patient record review. The vital status, cause of death, and need for readmission due to cardiovascular cause were recorded.

All of the study procedures were carried out in accordance with the ethical standards described in the Declaration of Helsinki. The patients provided written consent for the use of their clinical data for research purposes.

### 2.2. Statistical Analysis

The categorical data are expressed as frequencies and percentages and the continuous data as means with standard deviations (SDs) for the normally distributed variables, or medians and interquartile range (IQR) for the skewed variables. Departures from normality were evaluated using normal QQ-plots. Differences between the groups were obtained by the χ^2^ test or Fisher’s test for categorical variables, and by the *t*-test or Wilcoxon’s rank sum test for continuous variables, as appropriate.

The outcomes assessing survival were 30-day case-fatality, all-cause death from 30 days to 5 years, and a composite of all-cause death or cardiovascular readmission from 30 days to 5 years. The crude survival was analyzed using the Kaplan–Meier method and the survival rate of STEMI vs. NSTEMI compared using the log-rank test. Cox regression was used to determine the association between the electrocardiographic group and outcomes. The hazard ratios (HRs) are reported with 95% confidence intervals (CIs). The model was adjusted by using the enter method with covariates based on established predictors from prior knowledge [3,8,17]. The 30-day case-fatality multivariate analyses were performed using the following covariates: age; gender; previous cerebrovascular disease; previous MI or coronary artery bypass graft (CABG); cardiac arrest; LVEF on admission; glucose on admission; estimated glomerular filtration rate; and TIMI flow <3 after PCI or urgent CABG was performed. The long-term outcome models were adjusted by age, gender, diabetes mellitus, LVEF at discharge, and triple-vessel or left main disease. The probability values < 0.05 from two-sided tests were considered to indicate significance. All of the analyses were performed using the software IBM SPSS Statistics 24 (Chicago, IL, USA).

## 3. Results

During the study period, 4647 consecutive AMI patients (3407 STEMI and 1240 NSTEMI) were admitted, including 239 (5.1%) who developed AMI-CS (26.4% women): 190 STEMI-CS (5.6%) and 49 NSTEMI-CS (3.9%). The basal characteristics of the study population are shown in Table 1. The mean age was 69.7 (SD 11.6) years.

### 3.1. Clinical Profile

Although both of the groups were of similar age, the NSTEMI-CS patients had a higher prevalence of cardiovascular risk factors (former smoking, hypertension, and diabetes mellitus) and previous comorbidity, such as peripheral artery disease or end-stage chronic kidney disease. In addition, the NSTEMI-CS patients more frequently had a previous history of myocardial infarction, coronary revascularization (either percutaneous or surgical), and heart failure (Table 1).

The clinical presentation at admission also differed according to the pattern on the ECG (Table 1). The STEMI-CS patients had a lower systolic blood pressure and pH and higher blood lactate and glucose concentrations (i.e., were sicker), and had a larger infarct as measured by the peak creatine kinase-MB; the NSTEMI-CS patients presented with higher heart rates and lower glomerular filtration rates. However, the LVEF was similar between the groups, both on admission and at discharge. Cardiac complications were more frequent in the STEMI-CS patients, with significant differences in the case of mechanical complications (exclusive for STEMI) and atrioventricular block (Table 1). Cardiac surgery was performed in 50% of the STEMI patients with mechanical complications (17 patients).

Coronary angiography was performed in more than 90% of the patients from both groups. The main reason for not undergoing angiography was premature death after admission, before the procedure could be performed. Coronary anatomy and revascularization results are shown in Table 2. Briefly, angiography revealed a high prevalence of triple-vessel and left main disease in NSTEMI-CS; moreover, the left main was the most frequent infarct-related artery (IRA). No patients in this group had single-artery involvement. In contrast, almost one-third of the STEMI-CS patients had single-vessel disease, with the left anterior descending artery as the most frequent IRA. Regarding the initial coronary flow, the IRA was occluded (TIMI flow grades 0 and 1) in 84% of the STEMI-CS patients, but flow was preserved (TIMI grades 2 and 3) in 78% of the NSTEMI-CS patients.

Nearly 90% of all patients underwent revascularization, with similar rates in the two groups (Table 2). All of the STEMI-CS patients were treated with percutaneous coronary intervention (PCI) during the first 24 h after symptom onset, almost all of them with primary PCI (median 215 min). In the NSTEMI-CS group, 93% of the patients were revascularized with PCI in the first 24 h, and 4% were treated with emergency coronary artery bypass graft. Among those who underwent PCI, success was achieved more frequently in the NSTEMI-CS patients, with a TIMI flow grade 3 in most, whereas TIMI 3 patency was obtained in only two-thirds of the STEMI group (Table 2).

CS was managed with several invasive procedures in both groups. Mechanical ventilation was performed in almost two-thirds of all of the patients and an intra-aortic balloon pump (IABP) remained the most frequently used ventricular assistance device (45% of all of the patients), with similar rates between the groups. Extracorporeal membrane oxygenation (ECMO), available since 2019, was only used in the STEMI-CS patients (~5%). All of the therapies and procedures in the ICCU are shown in Table 3. The STEMI-CS patients were more frequently treated with newer P2Y12 inhibitors and antiarrhythmic drugs for ventricular tachycardia. Nevertheless, the NSTEMI-CS patients required more red blood cell transfusions and renal replacement therapy and received more treatment with inotropes, nitrates, diuretics, β-blockers, and statins. In contrast, the pharmacological treatment at discharge was similar between the two groups, highlighting only a greater use of nitrates and antiplatelet therapy with clopidogrel in the NSTEMI-CS patients (Appendix A).

### 3.2. Outcomes

We found some differences regarding the short- and long-term outcomes relative to the admission ECG pattern. From admission to day 30,131 patients died (54.8%). This acute-phase (30-day) mortality was higher in the STEMI-CS patients (59.5% vs. 36.7%, *p* = 0.004), as shown in the unadjusted survival curves in Figure 1A. After adjusting for the main confounding factors, STEMI-CS remained independently associated with 30-day mortality compared to NSTEMI, as shown in Table 4 and the Cox adjusted survival curves in Figure 2A.

After a median follow-up of 2.9 (IQR 1.4–5.0) years, 42 patients died (39.6% of 30-day survivors). The long-term mortality was higher in the NSTEMI-CS patients (64% vs. 40%, *p* = 0.049) than in the STEMI-CS patients. The main cause of death was non-cardiovascular disease, followed by heart failure and sudden death, without differences between the two groups. Causes of death among the hospital survivors are summarized in Appendix A. The cumulative mortality after the first year increased in the NSTEMI-CS patients, and the 30-day to 5-year mortality was higher in the NSTEMI-CS patients than in the STEMI-CS patients (mean survival 3.1 years [95% CI 2.4–3.8] vs. 3.8 [95% CI 3.4–4.2], *p* = 0.049), as depicted in the Kaplan–Meier curves in Figure 1B. However, after adjusting for clinical predictors, this association was no longer significant (Table 4 and Figure 2B). The overall 5-year mortality (including 30-day case-fatality) was similar in both groups, Figure 1C.

Regarding cardiovascular readmissions, among the 30-day survivors, 31 patients (28.7%) required hospitalization in the first 5 years. The causes of the first cardiovascular readmission among acute-phase survivors are summarized in Appendix A. Heart failure was the main cause of readmission in both of the groups, followed by acute coronary syndrome. Similarly, there was a non-significant trend of higher mortality or cardiovascular readmission in the NSTEMI-CS patients (mean 2.5 years [95% CI 1.8–3.2] vs. 3.0 [95% CI 2.5–3.5]). This association disappeared with adjustments (Table 4).

## 4. Discussion

In this prospective observational study, CS was ~30% more frequent with STEMI than NSTEMI. The STEMI-CS patients had larger myocardial infarctions and greater association with cardiac complications, including mechanical complications. The NSTEMI-CS patients had more severe coronary disease, with multivessel and left main artery involvement, and more associated comorbidities. Although the 30-day case-fatality was 38% higher in the STEMI-CS patients, the long-term prognosis was similar between the groups, due to a higher long-term mortality among the 30-day survivors with NSTEMI-CS.

Some of the studies have reported a higher prevalence of cardiovascular risk factors in NSTEMI patients [5,18] than in STEMI patients, which could probably explain the more extensive coronary artery disease. However, to the best of our knowledge, these differences have never been reported in the setting of AMI-CS. Multivessel coronary disease was reported in 62% of the NSTEMI patients in the recent CREDO registry [18], similar to a previous multicenter Spanish registry [5], though multivessel disease was present in 100% of our NSTEMI-CS patients. Moreover, the left main coronary artery was identified as the IRA in >40% of NSTEMI-CS patients in our series, 10-fold higher than in the CREDO STEMI patients [18]. In the setting of STEMI, these differences in the proportion of multivessel disease are not so pronounced, as it has been reported in 50–55% of all STEMI patients [5,18], whereas more than two-thirds of our STEMI-CS patients had more than one artery disease.

Short-term mortality was significantly higher in the STEMI-CS patients. As mentioned above, these patients had a worse initial hemodynamic situation, worse coronary flow after revascularization, and a higher rate of cardiac complications, which could explain the variance in mortality. The serious hemodynamic status of these patients prevented optimal myocardial perfusion, and patients with CS have worse results after PCI than patients with hemodynamic stability. Furthermore, suboptimal final coronary perfusion has been described as a strong predictor of in-hospital survival [3,19]. Our results concur with these data, as STEMI-CS had higher in-hospital case-fatality with worse angiographic results (TIMI flow) at the end of the procedure.

Nevertheless, a previous study published by Anderson [2] described higher in-hospital mortality in the NSTEMI-CS patients (40.8% vs. 33.1% in STEMI-CS). The main difference between both of the studies is the CS selection criteria in the STEMI patients. Anderson reported that 12.2% of STEMI patients had CS, with a 33% in-hospital mortality, whereas in our registry, CS developed in 5.6% of STEMI patients with 63.2% of in-hospital deaths. Moreover, there are some differences to highlight if we compare this previous study with our current project. In the American Registry, the patients with NSTEMI-CS were significantly older than the patients with STEMI-CS and underwent less revascularization than our NSTEMI-CS patients (35% vs. 87%). Both of the factors could be a determinant in mortality rates. Otherwise, in our study, there were no differences in age and revascularization between STEMI-CS and NSTEMI-CS patients, and both groups were comparable to each other. Nevertheless, our report only included those patients admitted to the ICCU, and we cannot rule out selection bias.

The patients with NSTEMI-CS had higher long-term mortality, and this is why the 5-year survival was similar in both groups. This may be related to the worse clinical profile of these patients, with more comorbidity that confers a worse prognosis. According to the higher frequency of extra-cardiological pathologies in these patients, the main cause of death during the follow-up was non-cardiovascular. The results from previously published studies have been variable, and included patients mainly without CS. In all of them, follow-up was shorter than in our current study. A study published in 2007 by Abbott et al. [4] did not show differences between STEMI and NSTEMI patients after 1 year of follow-up. Another paper, published in 2010 by Polonski et al. [6], described a worse unadjusted long-term prognosis in the NSTEMI patients but a worse prognosis in the STEMI patients, after adjusting for basal data. Finally, a study published in 2011 by Garcia-Garcia et al. [5] studied mid-term follow-up (6 to 12 months) and found a worse prognosis in NSTEMI patients.

Approximately one-third of the hospital survivors had readmission in the first 5 years of follow-up, with a 29% higher rate of readmission among the NSTEMI-CS patients. A high incidence of readmission in these patients was reported previously [20,21], ranging from 13.4% to 14.8% in the first 30 days from discharge. As described in these previous studies, the main cause of cardiovascular readmission in our cohort was heart failure due to ventricular dysfunction at discharge (moderate to severe in both groups), along with other comorbidities, such as diabetes or end-stage chronic kidney disease. Despite the multivessel disease involvement of most patients, readmissions for acute coronary syndrome were notably lower than for heart failure (18.8% vs. 68.8%). These results indicate that close follow-up at discharge by a specific heart failure unit should be considered to improve the prognosis of these patients.

## 5. Conclusions

In summary, the short-term mortality in AMI-CS varied, based on the electrocardiographic pattern present at admission. The STEMI-CS patients had ~40% higher 30-day case-fatality than the NSTEMI-CS patients. However, the NSTEMI-CS patients had greater long-term mortality, likely due to more comorbidities. As a result, the 5-year all-cause mortality was similar between the two groups. Regarding the clinical profile, the STEMI-CS patients had larger infarcts and more mechanical complications, whereas the NSTEMI-CS patients had more comorbidities and multi-vessel involvement.

## Figures and Tables

**Figure 1 jcm-11-03558-f001:**
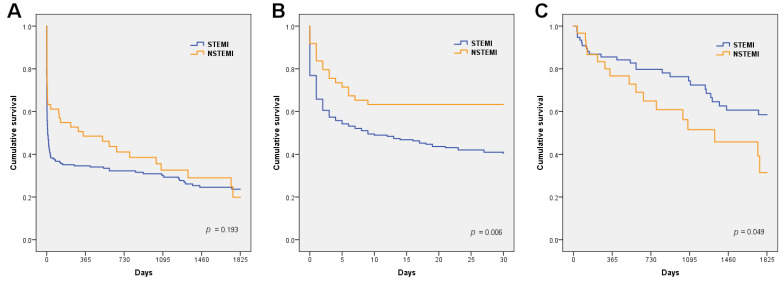
Kaplan–Meier curves according to ST-segment presentation showing non-adjusted cumulative survival. (**A**) All-cause death from admission to 5 years; (**B**) All-cause death in the first 30 days; (**C**) All-cause death from 30 days to 5 years.

**Figure 2 jcm-11-03558-f002:**
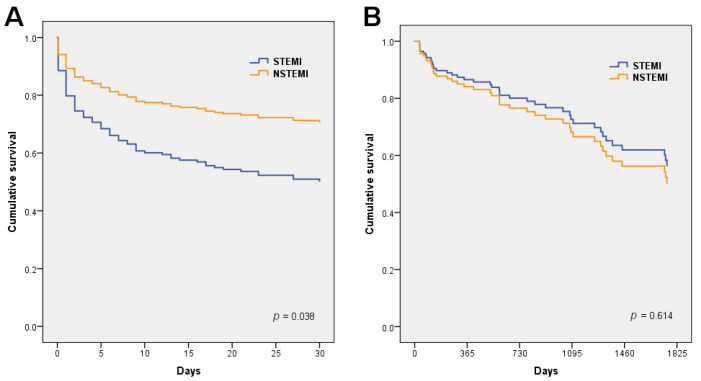
Cox-adjusted survival curves according to ST-segment presentation. (**A**) All-cause death in the first 30 days; (**B**) All-cause death from 30 days to 5 years.

**Table 1 jcm-11-03558-t001:** Baseline characteristics and clinical course based on presentation with or without ST-segment elevation.

	All Patients (*n* = 239)	STEMI Patients (*n* = 190)	NSTEMI Patients (*n* = 49)	*p* Value
Demographics				
Age, years	69.7 (11.6)	69.4 (11.9)	71.0 (10.1)	0.330
Gender, female	63 (26.4)	11 (22.4)	52 (27.4)	0.486
BMI, kg/m^2^	27.6 (4.7)	27.6 (4.9)	27.3 (4.2)	0.679
History				
Smoking	72 (30.1)	62 (32.6)	10 (20.4)	0.096
Former smoking	72 (30.1)	47 (24.7)	25 (51.0)	<0.001
Hypertension	159 (66.5)	119 (62.6)	40 (81.6)	0.012
Diabetes mellitus	95 (39.7)	69 (36.3)	26 (53.1)	0.033
Insulin treatment	35 (14.6)	21 (11.1)	14 (28.6)	0.002
Cerebrovascular disease	23 (9.6)	19 (10.0)	4 (8.2)	1
Peripheral artery disease	43 (18.0)	26 (13.7)	17 (34.7)	0.001
End-stage chronic kidney disease	4 (1.7)	1 (0.5)	3 (6.1)	0.028
Previous heart failure	14 (5.9)	7 (3.7)	7 (14.3)	0.005
Previous MI	41 (17.2)	19 (10.0)	22 (44.9)	<0.001
Q-wave	24 (10.0)	13 (6.8)	11 (22.4)	0.001
Non Q-wave	20 (8.4)	7 (3.7)	13 (26.5)	<0.001
Previous PCI	29 (12.1)	19 (10.0)	10 (20.4)	0.047
Previous CABG	9 (3.8)	4 (2.1)	5 (10.2)	0.020
Previous valvular surgery	1 (0.4)	1 (0.5)	0	1
Pacemaker carrier	4 (1.7)	4 (2.1)	0	0.584
Clinical presentation				
Systolic blood pressure, mmHg	98.4 (26.8)	96.5 (27.2)	105.6 (24.0)	0.034
Diastolic blood pressure, mmHg	60.8 (18.1)	60.4 (19.1)	62.5 (13.8)	0.387
Heart rate, bpm	92.8 (29.9)	90.6 (30.4)	101.1 (26.7)	0.029
Anterior infarct location (in STEMI)	117 (61.6)	117 (61.6)	-	-
LVEF on admission, %	32.1 (14.5)	32.6 (15.3)	30.4 (10.6)	0.231
CK-MB peak, ng/mL	229.0 (79.5–528.0)	278.7 (108.9–600.6)	85.0 (30.1–183.0)	<0.001
Hemoglobin, g/dL	12.2 (2.3)	12.3 (2.4)	11.8 (1.8)	0.150
Glucose, mg/dL	254.0 (117.9)	262.7 (120.1)	222.7 (104.9)	0.037
eGFR_CKD-EPI_, mL/min/1.73 m^2^	48.5 (24.6)	49.1 (23.1)	46.4 (29.6)	0.505
pH	7.24 (0.16)	7.22 (0.16)	7.30 (0.15)	0.004
Lactate, mmol/L	6.1 (5.1)	6.6 (5.1)	4.2 (4.7)	0.040
MI complications				
Cardiac arrest	102 (42.7)	83 (43.7)	19 (38.8)	0.536
Ventricular fibrillation	53 (22.2)	45 (23.7)	8 (16.3)	0.269
Sustained monomorphic ventricular tachycardia	45 (18.8)	38 (20.0)	7 (14.3)	0.362
Third degree atrioventricular block	49 (20.5)	45 (23.7)	4 (8.2)	0.017
Atrial fibrillation	61 (25.5)	49 (25.8)	12 (24.5)	0.852
Acute conduction disturbance	22 (9.2)	21 (11.1)	1 (2.0)	0.054
Any mechanical complication	34 (14.2)	34 (17.9)	0	<0.001
Free wall rupture	14 (5.9)	14 (7.4)	0	0.080
Ventricular septal rupture	16 (6.7)	16 (8.4)	0	0.048
Papillary muscle rupture	4 (1.7)	4 (2.1)	0	0.584
Discharge				
LVEF at discharge, %	34.6 (14.8)	34.3 (15.0)	35.7 (14.0)	0.551
Length of ACCU admission, days	4 (2–9)	4 (2–9)	5 (3–8)	0.174
Length of hospital admission, days	10 (2–22)	9 (2–21)	13 (4–22)	0.188
In-hospital mortality	138 (57.7)	120 (63.2)	18 (36.7)	0.001

STEMI, ST-segment elevation myocardial infarction; NSTEMI, non-ST-segment elevation myocardial infarction; BMI, body mass index; MI, myocardial infarction; PCI, percutaneous coronary intervention; CABG, coronary artery bypass graft; LVEF, left ventricular ejection fraction; CK-MB, creatine kinase-MB; eGFRCKD-EPI, estimated glomerular filtration rate by the Chronic Kidney Disease Epidemiology Collaboration formula; ACCU, acute cardiac care unit.

**Table 2 jcm-11-03558-t002:** Coronary anatomy and revascularization based on presentation with or without ST-segment elevation.

	All Patients (*n* = 239)	STEMI Patients (*n* = 190)	NSTEMI Patients (*n* = 49)	*p* Value
Catheterization lab data				
Coronary angiography	221 (92.5)	175 (92.1)	46 (93.9)	1
Main epicardial coronary arteries ≥70% stenosis				
0	2 (0.9)	2 (1.1)	0	1
1	52 (23.5)	52 (29.7)	0	<0.001
2	69 (31.2)	60 (34.3)	9 (19.6)	0.055
3	98 (44.3)	61 (34.9)	37 (80.4)	<0.001
Left main ≥ 50% stenosis	63 (28.5)	36 (20.6)	27 (58.7)	<0.001
Infarct-related artery				
None	2 (0.9)	2 (1.1)	0	1.000
Left main	46 (20.8)	27 (15.4)	19 (41.3)	<0.001
Left anterior descending	94 (42.5)	79 (45.1)	15 (32.6)	0.126
Ramus intermedius	3 (1.4)	1 (0.6)	2 (4.3)	0.111
Circumflex	25 (11.3)	19 (10.9)	6 (13.0)	0.677
Right coronary	48 (21.7)	45 (25.7)	3 (6.5)	0.004
Other	3 (1.4)	2 (1.1)	1 (2.2)	0.505
TIMI flow grade				
0	138 (62.4)	130 (74.3)	8 (17.4)	<0.001
1	19 (8.6)	17 (9.7)	2 (4.3)	0.377
2	23 (10.4)	15 (8.6)	8 (17.4)	0.081
3	41 (18.6)	13 (7.4)	28 (60.9)	<0.001
Revascularization data				
Revascularization	211 (88.3)	166 (87.4)	45 (91.8)	0.465
In the first 24 h	208 (98.6)	166 (100)	42 (93.3)	0.009
PCI	209 (87.4)	166 (87.4)	43 (87.8)	0.942
PCI in the first 24 h	206 (98.6)	166 (100)	40 (93.0)	0.008
Primary PCI	-	154 (92.8)	-	-
Symptom onset-to-balloon, min	-	215 (144–444)	-	-
TIMI flow grade after PCI				
0	17 (8.1)	16 (9.6)	1 (2.3)	0.206
1	11 (5.3)	11 (6.6)	0	0.125
2	27 (12.9)	25 (15.1)	2 (4.7)	0.078
3	154 (73.7)	114 (68.7)	40 (93.0)	0.001
PCI + staged CABG	3 (1.4)	1 (0.6)	2 (4.7)	0.108
Time to staged CABG, days	26 (3–63)	63 (63–63)	15 (3–26)	1
CABG	2 (0.8)	0	2 (4.1)	0.041
CABG in the first 24 h	2 (100)	-	2 (100)	-

Values are given as *n* (%) unless otherwise noted. STEMI, ST-segment elevation myocardial infarction; NSTEMI, non-ST-segment elevation myocardial infarction; TIMI, Thrombolysis In Myocardial Infarction; PCI, percutaneous coronary intervention; CABG, coronary artery bypass graft.

**Table 3 jcm-11-03558-t003:** Acute cardiovascular care unit procedures and medical therapies.

	All Patients (*n* = 239)	STEMI Patients (*n* = 190)	NSTEMI Patients (*n* = 49)	*p* Value
Procedures and treatments				
Mechanical ventilation				
Invasive	152 (63.6)	115 (60.5)	37 (75.5)	0.052
Non-invasive	30 (12.6)	22 (11.6)	8 (16.3)	0.371
Inotropes	222 (92.9)	173 (91.1)	49 (100)	0.027
Red blood cells transfusion	28 (11.7)	18 (9.5)	10 (20.4)	0.034
Temporary pacemaker	31 (13)	28 (14.7)	3 (6.1)	0.152
Pulmonary artery catheter	52 (21.8)	39 (20.5)	13 (26.5)	0.364
Renal replacement therapy	12 (5.0)	6 (3.2)	6 (12.2)	0.009
Ventricular assist devices				0.506
IABP	107 (44.8)	83 (43.7)	24 (49.0)	0.146
Impella CP	14 (5.9)	9 (4.7)	5 (10.2)	0.210
ECMO	9 (3.8)	9 (4.7)	0	0.027
Pharmacological treatment				
Dobutamine/dopamine	185 (77.4)	146 (76.8)	39 (79.5)	0.682
Epinephrine	40 (16.7)	33 (17.3)	7 (14.2)	0.266
Nitrates	101 (42.2)	74 (38.9)	27 (55.1)	0.041
Nitroprusside	14 (5.8)	12 (6.3)	2 (4.0)	0.741
Other vasodilators	37 (15.4)	31 (16.3)	6 (12.2)	0.482
Diuretics	152 (63.5)	114 (60.0)	38 (77.5)	0.023
Aspirin	200 (83.6)	156 (82.1)	44 (89.7)	0.194
P2Y12 inhibitors	178 (74.4)	137 (72.1)	41 (83.6)	0.098
Clopidogrel	160 (66.9)	121 (63.6)	39 (79.5)	0.035
Prasugrel	27 (11.2)	24 (12.6)	3 (6.1)	0.310
Ticagrelor	8 (3.3)	6 (3.1)	2 (4.0)	0.669
Glycoprotein IIb/IIIa inhibitors	30 (12.5)	27 (14.2)	3 (6.1)	0.152
Any heparin	187 (78.2)	146 (76.8)	41 (83.6)	0.301
Low-molecular-weight heparin	70 (29.2)	56 (29.4)	14 (28.5)	0.902
Unfractionated heparin	159 (66.5)	124 (65.2)	35 (71.4)	0.415
Amiodarone	66 (27.6)	51 (26.8)	15 (30.6)	0.599
Lidocaine	25 (10.4)	25 (13.1)	0	0.003
Other antiarrhythmic drugs	3 (1.2)	3 (1.5)	0	1
Digoxin	17 (7.1)	14 (7.3)	3 (6.1)	1
ACEIs/ARBs	95 (39.7)	79 (41.5)	16 (32.6)	0.255
β-Blockers	52 (21.7)	35 (18.4)	17 (34.6)	0.014
Calcium channel blockers	18 (7.5)	17 (8.9)	1 (2.0)	0.133
Statins	132 (55.2)	97 (51.0)	35 (71.4)	0.011

Values are given as *n* (%). STEMI, ST-segment elevation myocardial infarction; NSTEMI, non-ST-segment elevation myocardial infarction; IABP, intra-aortic balloon pump; ECMO, extracorporeal membrane oxygenation; ACEI, angiotensin-converting enzyme inhibitor; ARB, angiotensin receptor blocker.

**Table 4 jcm-11-03558-t004:** Cox regression analyses of outcomes.

	**Univariable**			**Multivariable**		
30-day all-cause death (*n* = 239)						
	HR	95% CI	*p* value	HR	95% CI	*p* value
STEMI *	1.91	1.16–3.14	0.011	1.99	1.04–3.83	0.038
Age	1.04	1.02–1.05	<0.001	1.05	1.02–1.07	<0.001
Gender, female	1.31	0.91–1.91	0.151	1.35	0.82–2.22	0.247
Cerebrovascular disease	1.40	0.93–236	0.210	1.15	0.61–2.15	0.667
Previous MI or CABG	0.77	0.49–1.23	0.282	1.07	0.58–1.96	0.825
Cardiac arrest	2.45	1.73–3.47	<0.001	2.54	1.62–4.00	<0.001
LVEF on admission	0.98	0.96–0.99	<0.001	0.98	0.97–1.00	0.045
Glucose on admission	1.00	1.00–1.00	0.007	1.00	1.00–1.00	0.593
eGFR_CKD-EPI_ on admission	0.99	0.98–1.00	0.003	0.99	0.98–1.00	0.047
TIMI < 3 after PCI or urgent CABG	0.41	0.28–0.60	<0.001	0.47	0.30–0.75	0.001
30-day to 5-year all-cause death (*n* = 106)						
	HR	95% CI	*p* value	HR	95% CI	*p* value
STEMI *	0.54	0.29–1.01	0.052	0.83	0.41–1.70	0.614
Age	1.03	1.00–1.06	0.035	1.02	0.99–1.05	0.230
Gender, female	0.53	0.23–1.27	0.157	0.63	0.26–1.54	0.313
Diabetes mellitus	2.70	1.43–5.08	0.002	1.95	0.99–3.83	0.052
LVEF at discharge	0.96	0.94–0.99	0.002	0.97	0.94–1.00	0.026
Triple-vessel or left main disease	2.07	1.08–3.96	0.028	1.40	0.69–2.84	0.356
30-day to 5-year all-cause death or cardiovascular readmission (*n* = 106)						
	HR	95% CI	*p* value	HR	95% CI	*p* value
STEMI *	0.70	0.40–1.22	0.210	1.11	0.59–2.09	0.756
Age	1.04	1.01–1.06	0.012	1.03	1.00–1.06	0.059
Gender, female	0.79	0.40–1.57	0.500	0.79	0.39–1.61	0.515
Diabetes mellitus	2.35	1.37–4.00	0.002	1.85	1.04–3.30	0.035
LVEF at discharge	0.97	0.95–1.00	0.016	0.98	0.96–1.00	0.069
Triple-vessel or left main disease	1.86	1.07–3.24	0.028	1.30	0.69–2.45	0.415

* NSTEMI as reference. HR, hazard ratio; CI, confidence interval; MI, myocardial infarction; CABG, coronary artery bypass graft; LVEF, left ventricular ejection fraction; eGFR_CKD-EPI_, estimated glomerular filtration rate by the Chronic Kidney Disease Epidemiology Collaboration formula; TIMI, Thrombolysis In Myocardial Infarction grade; PCI, percutaneous coronary intervention.

## Data Availability

Results are contained within the article.

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
