# Peer review of "Non-STEMI vs. STEMI Cardiogenic Shock: Clinical Profile and Long-Term Outcomes"

_jcm, 2022, doi:10.3390/jcm11123558_

Round 1

Reviewer 1 Report

The study is very interesting for me, because there are various debates about that: which type of the myocardial infarction is more unfavorable prognostically? The authors obtained interesting data on the prevalence of cardiogenic shock, analyzed its risk factors in different types of myocardial infarction, and traced the long-term prognosis. With the same parameters of the ejection fraction, different predictors of unfavorable prognosis were obtained. It is important, that the study analyzed all cases of myocardial infarction over a certain period of time, which makes the results more accurate. It included a large number of patients.

In my opinion, this study is a significant contribution to the study of the pathogenesis of cardiogenic shock in myocardial infarction. Mortality in this complication of myocardial infarction remains high in all countries.
A better understanding of its pathogenesis allows for timely prevention and building an emergency care algorithm
This is very relevant and interesting, for me it was also interesting because a sample of patients was made in one region, similar in population to the Republic of Karelia, the region where I live and the statistics of myocardial infarction complications are very similar
The topic is not original, but it is extremely important for a practicing cardiologist, therefore it is undoubtedly of interest
The study of risk factors for cardiogenic shock depending on the type of infarction is the main difference from previous studies. Previously, most publications and registries analyzed the overall prognosis for different types of myocardial infarction, without such an in-depth analysis of the prognosis for cardiogenic shock
The article is well written and easy to read.
Conclusions follow from the results of the study and answer the main question posed as the purpose of the work

Author Response

  • Thank you very much for the reviewer comments. It is a good summary of our results.
  • The reviewer recognized the effort of the authors in increasing the knowledge of the pathogenesis of cardiogenic shock in myocardial infarction. This is one of the main strengths of the manuscript.

Reviewer 2 Report

Excellent prospective observational study examining short and long term clinical outcomes in STEMI vs NSTEMI patients presenting with cardiogenic shock.  This type of risk stratification of cardiogenic shock has not been previously described.  The authors findings are consistent and complementary to existing literature indicating the distribution of short vs long-term outcomes in STEMI vs NSTEMI patients.  

This is prospective observational study examining short and long term clinical outcomes in STEMI vs NSTEMI patients presenting with cardiogenic shock.  
This type of risk stratification of cardiogenic shock has not been previously described and will be of interest to clinicians managing these types of patients. The authors findings are consistent and complementary to existing literature indicating the distribution of short vs long-term outcomes in STEMI vs NSTEMI patients.  
The paper is well written with clear and easy to read text.  
The authors conclusions are consistent with the observational data presented and does provide useful insight on the short and long term clinical outcomes in STEMI vs NSTEMI patients presenting with cardiogenic shock. 

Author Response

- We appreciate the reviewer's comment and agree this is an important issue with a new stratification of cardiogenic shock that has not been previously described. 

- Thank you very much for the review. It is a good summary of our manuscript. 

Reviewer 3 Report

The paper "Non-STEMI vs. STEMI cardiogenic shock: clinical profile and long-term outcomes" was reviewed. The paper was well written and the analysis was appropriate. I have only 2 comments.

The proportion of STEMI seems to be larger than that of NSTEMI. it is stated that the diagnosis is based on universal definition, but is it registered only for Type 1 MI or does it include other types of MI?

Please provide details on how many cases of repair surgery were performed for mechanical complications.

Author Response

We appreciated general comments of reviewer 3 and we answer the 2 questions he/she made.

  1. The proportion of STEMI seems to be larger than that of NSTEMI. it is stated that the diagnosis is based on universal definition, but is it registered only for Type 1 MI or does it include other types of MI?

- Thank you very much for the reviewer's comment. For this registry, only type 1 AMI was considered. We included a sentence on page 2, in "Material and methods" section with this information.

  1. Please provide details on how many cases of repair surgery were performed for mechanical complications

- Attending to the reviewer's suggestion, on page 4, in the results section this sentence has been included: "Cardiac surgery was performed in 50% of STEMI patients with mechanical complications (17 patients)".